# Physicochemical, Microbiological and Functional Properties of Camelina Meal Fermented in Solid-State Using Food Grade *Aspergillus* Fungi

**Oladapo Oluwaseye Olukomaiya [1]**, **W. Chrishanthi Fernando [1]**, **Ram Mereddy [2]**, **Xiuhua Li [3]** and **Yasmina Sultanbawa [1,*]**

[1] Centre for Nutrition and Food Sciences, Queensland Alliance for Agriculture and Food Innovation (QAAFI), The University of Queensland, Archerfield BC, Coopers Plains, QLD 4108, Australia; o.olukomaiya@uqconnect.edu.au (O.O.O.); chrishanthif2@gmail.com (W.C.F.)

[2] Queensland Department of Agriculture and Fisheries, Archerfield BC, Coopers Plains, QLD 4108, Australia; ram.mereddy@daf.qld.gov.au

[3] Poultry Science Unit, School of Agriculture and Food Sciences, The University of Queensland, Gatton, QLD 4343, Australia; x.li1@uq.edu.au

* Correspondence: y.sultanbawa@uq.edu.au

**Abstract:** Camelina meal (CAM) was fermented in solid-state using food grade *Aspergillus* fungi (*A. sojae*, *A. ficuum* and their co-cultures), and the physicochemical composition, microbiological and functional properties were investigated. SSF increased the starch contents but reduced ($p < 0.05$) the contents of soluble carbohydrate. The microbiological counts of the fermented meals were higher ($p < 0.05$) than that of the unfermented CAM. Phytic acid content reduced ($p < 0.05$) in the fermented meals. SSF reduced the protein molecular weight and colour attributes of CAM. The fermented camelina meals had increased ($p < 0.05$) bulk density and swelling capacity but reduced ($p < 0.05$) water absorption capacity. Thus, the study indicated that SSF with *A. sojae*, *A. ficuum* and their co-cultures influenced the physicochemical, microbiological and functional properties of CAM. There is potential for the development of value-added novel food and feed products from solid-state fermented camelina meal.

**Keywords:** proximate composition; *Aspergillus* fungi; solid-state fermentation; camelina meal; protein molecular distribution

## 1. Introduction

Camelina (*Camelina sativa* L), also known as gold of pleasure or false flax is an oilseed crop which originates from Northern Europe and Central Asia and belongs to the *Brassicaceae* family. It is mainly cultivated for its oil content useful for the biofuel industries [1]. Camelina oil contains about 45% polyunsaturated fatty acid (PUFA), 35% monounsaturated fatty acid, 10% saturated fatty acid and less than 10% free fatty acids in addition to tocopherols, sterols, terpenes and volatiles [2]. Camelina meal (CAM) recovered as a by-product of oil extraction from camelina seeds is useful in feeding livestock and aquatic species and, has gained popularity in recent time. It has moderate levels of amino acid, minerals and vitamins. CAM contains about 35.2% crude protein (CP), high PUFA concentration and antioxidant compounds which are beneficial in improving meat quality [3]. Besides, interest in protein-rich material is very important as the knowledge of their functionality will influence their use in food production. It has also been revealed that camelina protein has desirable functional properties that can make it competitive with soy protein [4]. However, the use of CAM in food and feed production is restricted due to the presence of anti-nutritional factors (ANFs). CAM contains ANFs such as total

glucosinolates of about 15.2–24.6 mmol/g, 25.4–32.3 g/kg phytic acid, 1.58–2.93 g/kg sinapine and 1.92 to 4.39 g/kg condensed tannins content [5]. For instance, these ANFs are known to adversely affect growth performance and nutrient digestibility in non-ruminant animals [6]. Recently, strategies such as solvent-extraction [7] and inclusion of exogenous enzymes [8] have been attempted to improve the nutritional value of CAM for use in animal feed. As value-addition is one of the critical issues in managing wastes from the agro and food processing industry [9], it is important to explore sustainable ways of processing agro-industrial by-products/wastes, among which is solid-state fermentation (SSF). SSF is a viable technique useful for enhancing the nutrient quality of agro-industrial by-products and for producing high-value products of commercial importance such as enzymes, organic acids among many others [10–12]. The potential elimination of ANFs by SSF is important for increasing the economic value and for making it possible for CAM to be well utilized as animal feed or food products in the food and feed industries. The beneficial effect of SSF in improving the nutritive value and bioactivity of various oilseed meals such as rapeseed meal [10], canola meal [13], soybean meal [14] and cottonseed meal [15] using different types of microorganisms have been previously reported. In an earlier study, CAM treated with *Saccharomyces cerevisiae* (yeast) was found to contain reduced phytic acid content compared to the unfermented control [16]. Filamentous fungi such as *Aspergillus sojae* and *Aspergillus ficuum* are important food-grade fungi which have been used in SSF technologies for several years, especially in the production of traditional Asian fermented foods [17]. They have been certified as safe with regards to their epidemiological features and GRAS (Generally Regarded as Safe) status in the food industry [18,19]. Furthermore, *A. sojae* and *A. ficuum* have been used in decreasing the levels of ANFs, improving nutrient composition of food and agro-industrial by products, and for enzyme production [20–22]. Although, the use of CAM in food and feed products seems beneficial, most studies have been conducted on extraction and characterization of camelina protein [4,23,24] as well as in the evaluation of phenolic profiles and antioxidant activity [25]. To the best of our knowledge, there is limited information on the use of CAM as a fermentation substrate in SSF systems. Therefore, the purpose of the present study was to confirm the suitability of CAM as a substrate for SSF using *A. sojae*, *A. ficuum* and their co-cultures in creating value-added novel products. The products obtained after SSF were assessed for their proximate composition, anti-nutritional factors, microbiological and functional properties. In addition, the fibre fractions, protein molecular distribution and colour attributes of the products were evaluated.

## 2. Materials and Methods

### 2.1. Materials

Expeller-extracted camelina meal (CAM) was purchased from a commercial feed mill (Cobbett Pty. Ltd., New South Wales, Australia) and stored at 4 °C. All the reagents and media used in the study were of analytical grade. Folin-Ciocalteu phenol's reagent, gallic acid, phytic acid (Inositol hexaphosphoric acid) dodecasodium salt, sinigrin hydrate and 5-sulfosalicylic acid dihydrate were purchased from Sigma-Aldrich (Saint-Louis, MO, USA). Sodium tetrachloropalladate (II) was purchased from Fluorochem Ltd., Glossop, UK.

### 2.2. Preparation of Samples

2.2.1. Preparation of Fungi Inocula

Lyophilized cultures of fungal strains, *Aspergillus sojae* (ATCC 9362) and *Aspergillus ficuum* (ATCC 66876), were purchased from the American Type Culture Collection (ATCC, Manassas, VA, USA). After activation, *A. sojae* was maintained on PDA (potato dextrose agar; Oxoid Ltd., Basingstoke, UK) slants at 37 °C while *A. ficuum* was maintained on CDA (czapek dox agar; Oxoid Ltd., Basingstoke, UK) slants at 25 °C for 7 days. Spores were harvested by gently washing with 0.1% Tween 80 (Ajax Chemicals, Australia). Spore counts were determined using a colony counter (Stuart Scientific, UK) and working

suspensions were prepared at $10^7$ spores/mL. All microbiological procedures were conducted under aseptic conditions.

### 2.2.2. Solid-State Fermentation

Solid-state fermentation was conducted according to a previous protocol [17]. Each SSF set-up was prepared in triplicate in 500-mL Erlenmeyer flasks. Moisture content of the substrates (150 g) was adjusted by 45% with reverse osmosis (RO) water before sterilization by autoclaving (Sabac autoclave Model T62, Australia) at 121 °C for 15 min. The cooled CAM samples were inoculated with spore suspension and the mixture was incubated at 30 °C for 7 days. Autoclaved and unfermented CAM served as the control. The inoculum of *A. sojae* alone or *A. ficuum* alone was used for monoculture SSF while a combination of both strains (ratio 1:1) was used for co-culture SSF.

### 2.3. pH, Total Titratable Acidity and Colour Attributes

The pH and total titratable acidity of unfermented and fermented samples were determined according to a previous method [26]. The pH values were determined using a pH meter (PHM 210, Meterlab, Radiometer Analytical SAS, Copenhagen, Denmark). Total titratable acidity was determined on a 10 g of sample mixed with 90 mL RO water and the suspension was titrated against 0.1 mol /L NaOH. The total titratable acidity was expressed as the amount (mL) of 0.1 mol /L NaOH to attain a pH value of 8.2. Colour attributes of samples were determined according to a previous protocol [17] using Minolta colorimeter (Model CR-400, Konica Minolta Co Ltd., Osaka, Japan). Prior to the analysis, the instrument was calibrated with a standard white tile (Y = 88.2, x = 0.3158 and y = 0.3229). The colour of the samples was determined in terms of CIE *L*, *a*, *b* values of L (lightness-darkness), a (redness-greenness) and b (yellowness-blueness).

### 2.4. Proximate Analysis

Proximate analysis was carried out at the School of Agriculture and Food Sciences, The University of Queensland, Brisbane. The unfermented and fermented samples were analyzed in triplicate according to standard methods [27] for dry matter (method 925.10), crude protein (method 990.03), crude fat by Soxhlet extraction (method 960.39), crude ash (method 923.03) and starch (method 996.11). Soluble carbohydrate (glucose) was measured by enzymatic method [28]. Calcium and phosphorus concentrations were determined using inductively coupled plasma atomic emission spectroscopy (ICP-AES [29]).

### 2.5. Anti-Nutritional Factors

### 2.5.1. Phytic Acid

Phytic acid content of samples was analyzed according to the modified colorimetric method [30]. Approximately, 1 g of sample was extracted with 20 mL 2.4% HCl with a rotary shaker for 16 h at room temperature. The extracts were centrifuged at 4000× *g* for 10 min at 10 °C. The supernatant was mixed with 2 g NaCl for 20 min. After resting for 60 min at 4 °C, tubes were centrifuged at 4000× *g* for 20 min at 10 °C. Exactly 100 μL of the supernatant, 1900 μL RO water, 300 μL Wades reagent were mixed and immediately measured against RO water at 500 nm with a spectrophotometer (Thermo Fisher Scientific GeneSys 20, Victoria, Australia). Phytic acid (Inositol hexaphosphoric acid) dodecasodium salt was used as standard. Phytic acid content was calculated from the standard calibration curve and expressed as mg PA/g sample.

### 2.5.2. Total Glucosinolates

Total glucosinolates of samples were measured by the modified sodium tetrachloropalladate spectrophotometric method using a sinigrin hydrate calibration curve (0.03125–1 mmol/L) according to previously described protocols [31,32]. In brief, methanolic extracts were prepared by homogenizing

0.1 g of sample with 80% methanol. Into 100 μL of extract, 0.3 mL distilled water and 3mL of 2mmol/L sodium tetrachloropalladate were added. The mixture was incubated for 1 h and absorbance read at 420 nm. Results were calculated as sinigrin equivalents (SE) and expressed in μmol SE/g sample.

### 2.6. Total Phenolic Contents

Total phenolic contents of samples were determined using Folin-Ciocalteu's reagent with gallic acid as the standard [33]. Briefly, 0.5 g of sample was extracted twice with 10 mL of 80% methanol for 10–15 min. An aliquot of 25 μL extract was mixed with 125 μL of 10% Folin-Ciocalteu reagent, 125 μL of 0.7 mol/L sodium carbonate and allowed to incubate in the dark for 30 min at room temperature. Absorbance was measured at 750 nm using an automated microplate reader (Tecan infinite F200 96 well plate reader, Austria). Total phenolic contents were calculated as gallic acid equivalents (GAE) from the calibration curve and expressed as mg GAE/g sample.

### 2.7. Fibre Fractions

Neutral detergent fibre was determined according to the method described by Van Soest [34,35]. Neutral detergent fibre was measured as the insoluble part of a neutral detergent solution. Acid detergent fibre is the portion of insoluble components in the acid detergent solution and was determined according to the method of Van Soest [34,35]. Hemicellulose was calculated by the difference between neutral detergent fibre and acid detergent fibre contents. Lignin content in acid detergent fibre was determined according to the method of Van Soest [34,35], which defined it as the insoluble lignin fraction in 72% sulphuric acid. Cellulose was calculated by the difference between Acid detergent fibre and lignin.

### 2.8. In Vitro Enzyme Protein Digestion (IVPD)

In vitro enzyme protein digestion (IVPD) was determined according to a previously reported method [36]. One gram of sample was suspended in 5 mL distilled water and incubated at 40 °C for 30 min. Then, 0.35 mL 1 mol/L HCl was added and pH was adjusted between 2.7–2.9. One mL pepsin solution (containing 10 mg pepsin) was added and mixture was incubated at 40 °C for 1 h. Into the mixture, 0.30 mL of 1 mol/L NaOH was later added, pH adjusted between 5.9–6.10, then 1 mL of 5% pancreatin solution was added followed by incubation at 40 °C for 2 h. One mL of 40% 5-sulfosalicylic acid dihydrate was added, vortexing of the mixture was done and allowed to rest for 30 min. At the end of 30 min, the mixture was centrifuged at 2500× *g* for 20 min, supernatant decanted and residue was mixed again with 40% 5-sulfosalicylic acid dihydrate. Sample tubes were filled with 10 mL of 95% ethanol, mixture was vortexed, centrifuged twice (at 2500× *g* for 20 min) and supernatant decanted. Ten mL of acetone was added, mixed with the residue and centrifuged at 2500× *g* for 20 min (repeated twice). The supernatant was finally decanted. The final residue was dried, weighed and analyzed for nitrogen content. The IVPD (%) was calculated as:

$$IVPD\ (\%) = \frac{(I - F)}{I} \times 100$$

where, *I* = protein content of sample before digestion, *F* = protein content of sample after digestion.

### 2.9. Microbiological Analysis

Microbiological analysis was carried out according to standard plate methods [17,37]. Total bacteria in the samples were enumerated on standard plate count agar incubated at 30 °C for 72 h. Lactic acid bacteria (LAB) were enumerated on MRS (de Man, Rogosa and Sharpe; Oxoid Ltd., Basingstoke, UK) agar incubated anaerobically at 30 °C for 72 h. Molds and yeasts were enumerated on PDA (potato dextrose agar; Oxoid Ltd., Basingstoke, UK) incubated at 25 °C for 96 h. *Salmonella* was determined at Symbio Alliance, Queensland, Australia (a National Association of Testing Authorities (NATA) accredited laboratory) using the horizontal method (procedure M16.1 AS 5013.10).

### 2.10. Sodium Dodecyl Sulfate-Polyacrylamide Gel Electrophoresis (SDS-PAGE)

The analysis of protein component degradation by SDS-PAGE was conducted at the Protein Expression Facility of the University of Queensland, Brisbane according to a previous procedure [17]. Peptide extraction from crude milled samples was done using a Precellys Evolution homogenizer (Thermo Fisher Scientific, Victoria, Australia) with either 1.4 mm ceramic beads or 0.5 mm glass beads. Briefly, 0.05 g of milled sample was added to a 2 mL screw top tubes containing the 1.4 mm ceramic or 0.5 mm glass beads. One mL of lysis buffer (20 mmol/L Tris-HCl, with or without 5 mmol/L Dithiothreitol (DTT), 0.1% SDS and Roche complete protease inhibitor) was added to each tube and homogenized at 6500 rpm. Samples were kept on ice throughout the homogenisation process. Samples were centrifuged at 20,000× *g* for 5 min at 4 °C after homogenisation to collect the soluble fraction for SDS-PAGE analysis.

### 2.11. Functional Properties

#### 2.11.1. Bulk Density

Bulk density was determined according to the method described in a previous study [38]. Into a 10 mL graduated measuring cylinder, 2 g of sample was placed and the base of the cylinder was gently tapped severally to achieve a constant volume. The bulk density (g/mL) was expressed as weight of flour (g) per flour volume (mL).

#### 2.11.2. Water Absorption Capacity

Water absorption capacity was determined at room temperature based on a previously reported method [39]. About 1 g of sample was homogenized with 10 mL RO water and centrifuged at 3000× *g* for 30 min. The water absorption capacity was expressed as percentage increase in the weight of the sample.

#### 2.11.3. Oil Absorption Capacity

Oil absorption capacity was determined according to a centrifugal method [40]. About 0.5 g of sample was homogenized with 3 mL of canola oil and centrifuged at 3000× *g* for 30 min. The oil absorption capacity was expressed as percentage increase in the weight of the sample.

#### 2.11.4. Swelling Index

The method described in an earlier study [41] was used for the determination of swelling index. About 10 g of sample was placed in a measuring cylinder and 50 mL of RO water was added. After standing for 4 h, swelling index was determined as the ratio of the swollen volume to the initial volume.

#### 2.11.5. Swelling Capacity

Swelling capacity was determined according to a previously reported method [42]. A 100 mL graduated cylinder was filled with sample to 10 mL mark and RO water was added to make a total volume of 50 mL. The graduated cylinder was tightly covered, inverted and shaken properly. The suspension was again inverted after 2 min and allowed to rest for another 8 min. The final volume occupied by the sample was measured after the 8th min.

#### 2.11.6. Dispersibility

Dispersibility was determined according to a previously reported method [43]. About 1 g of sample was placed in a 20 mL measuring cylinder and RO water was added to attain the 10 mL mark. The mixture was stirred vigorously and allowed to rest for 3 h after which the volume of settled particle was measured.

### 2.12. Statistical Analysis

All experiments were conducted in triplicate and all measurements were done in triplicate. The results are expressed as mean ± standard deviation. Statistical analysis was done using IBM SPSS

Statistics for Windows, V.25 (IBM Corp., Armonk, NY, USA). The data obtained were analyzed for mean differences with one-way analysis of variance (ANOVA) using Tukey's procedure with a significance level of $p < 0.05$ as well as assumptions of normality and homogeneity of variances.

## 3. Results and Discussion

### 3.1. Proximate Composition

The proximate composition of unfermented and solid-state fermented camelina meals is presented in Table 1. The crude protein (CP) content in UCAM appear to slightly increase by 1.46% in ASCAM, 2.00% in AFCAM and 2.11% in COCAM. The margin of increase in CP content in the present study is lower compared to earlier reports which found 8.1% and 16.5% increase in CP content of canola meal fermented with food grade *Aspergillus sojae* and *Aspergillus ficuum*, respectively [11] and a 2.4% increase in CP content of canola meal fermented with *Lactobacillus salivarius* [13]. The initial crude fat content in UCAM (15.63%) also seem to decrease to 15.47% in ASCAM, 15.31% in COCAM and 15.18% in AFCAM. The result is consistent with a previous report [44] which showed a decrease in crude fat content of canola meal fermented with *Saccharomyces cerevisiae*. The effects of fermentation on nutritional profile has produced varying results probably due to different experimental designs, duration of study, variation in the initial protein or amino acid profile of substrates, differences in fermentation conditions [12]. It is possible that most of the effects of SSF on crude protein, crude fat and other nutritional components may not reflect actual changes but relative changes due to loss of dry matter as a result of microorganisms hydrolyzing and metabolizing carbohydrates and fats as carbon and energy sources during fermentation [10,17,45]. Soluble carbohydrate (glucose) content of 8.71% in UCAM decreased ($p < 0.05$) to 6.46% in COCAM, 6.35% in ASCAM and 6.07% in AFCAM. The decrease in soluble carbohydrate may be attributed to the use of glucose as a source of energy by the fungi [17,46]. The starch content of 0.18% in UCAM increased ($p < 0.05$) to 1.13% in ASCAM, 0.94% in AFCAM and 1.03% in COCAM. This result is in line with an earlier finding [47] which reported an increase in the starch content of soy flours after fermentation. However, fermented flours are typically known to have lower starch and higher sugar contents [37,48], thus, more studies are needed to investigate the variations in results.

**Table 1.** Proximate composition of unfermented and solid-state fermented camelina meals (DM basis).

| Parameters | UCAM | ASCAM | AFCAM | COCAM |
|---|---|---|---|---|
| Crude protein (%) | 37.03 ± 0.06 [a] | 37.57 ± 0.06 [a] | 37.77 ± 0.06 [a] | 37.81 ± 0.01 [a] |
| Crude fat (%) | 15.63 ± 0.04 [a] | 15.47 ± 0.07 [a] | 15.18 ± 0.10 [a] | 15.31 ± 0.03 [a] |
| Crude ash (%) | 5.92 ± 0.19 [a] | 5.90 ± 0.14 [a] | 5.90 ± 0.06 [a] | 5.81 ± 0.13 [a] |
| Soluble carbohydrate (glucose, %) | 8.71 ± 0.02 [a] | 6.35 ± 0.03 [c] | 6.07 ± 0.05 [d] | 6.46 ± 0.03 [b] |
| Starch (%) | 0.18 ± 0.02 [c] | 1.13 ± 0.06 [a] | 0.94 ± 0.03 [b] | 1.03 ± 0.06 [ab] |
| Calcium (%) | 0.22 ± 0.10 [a] | 0.31 ± 0.26 [a] | 0.24 ± 0.16 [a] | 0.21 ± 0.10 [a] |
| Phosphorus (%) | 0.77 ± 0.13 [a] | 0.86 ± 0.28 [a] | 0.82 ± 0.21 [a] | 0.78 ± 0.14 [a] |

Values are given as mean ± standard deviation (n = 3). Means in the same row having different superscripts (a, b, c, d) are significantly different ($p < 0.05$). UCAM: unfermented camelina meal; ASCAM: *Aspergillus sojae* fermented camelina meal; AFCAM: *Aspergillus ficuum* fermented camelina meal; COCAM: co-culture fermented camelina meal.

### 3.2. pH, Total Titratable Acidity and Colour Attributes

There was a reduction ($p < 0.05$) in pH values from 6.07 (UCAM) to 6.03 (ASCAM), 5.87 (AFCAM) and 5.94 (COCAM), with AFCAM having the lowest pH value (Table 2). This result agrees with the reports of other studies [11,13,16] which also observed reduction in the pH of canola meal fermented with *Lactobacillus salivarius* and CAM fermented with *Saccharomyces cerevisiae*. A decline in pH suggests increase in organic acids produced by the microorganisms during SSF [49] or breakdown of carbohydrates into organic acids [50]. Total titratable acidity gradually increased ($p < 0.05$) from the initial value of 0.45 (UCAM) to 0.95, 1.13, 1.26 in AFCAM, COCAM and ASCAM, respectively but the level of increase did not differ ($p > 0.05$) among the fermented meals. In accordance with the above

results, increase in total titratable acidity values can discourage the rapid multiplication of unwanted microorganisms that may want to induce poor fermentation [51]. Colour attributes (L*, a* and b*) significantly reduced ($p < 0.05$) after SSF with the unfermented meal having higher colour values than the solid-state fermented camelina meals (Table 2). The result is in agreement with an earlier report [52] which found that fermentation reduced L*, a* and b* values of tarhana (a wheat flour–yoghurt mixture). The reduction in colour attributes of the fermented meals may be due to thermal degradation of colour pigments or effect of decrease in pH during SSF causing discolouration of the fermented meals [17,46].

**Table 2.** pH, total titratable acidity, fibre fractions (on DM basis) and colour attributes (L, a and b values) of unfermented and solid-state fermented camelina meals.

| Parameters | UCAM | ASCAM | AFCAM | COCAM | CV (%) |
|---|---|---|---|---|---|
| pH | 6.07 ± 0.01 [a] | 6.03 ± 0.01 [b] | 5.87 ± 0.01 [d] | 5.94 ± 0.01 [c] | 0.17 |
| Total titratable acidity | 0.45 ± 0.14 [b] | 1.26 ± 0.12 [a] | 0.95 ± 0.09 [a] | 1.13 ± 0.31 [a] | 19.39 |
| Neutral detergent fibre (%) | 33.49 ± 2.41 [a] | 36.94 ± 2.51 [a] | 38.48 ± 2.84 [a] | 40.36 ± 3.44 [a] | 7.47 |
| Acid detergent fibre (%) | 16.86 ± 0.04 [b] | 20.76 ± 0.67 [a] | 21.47 ± 1.98 [a] | 19.96 ± 0.76 [a] | 4.12 |
| Cellulose (%) | 12.25 ± 0.51 [b] | 14.79 ± 0.64 [a] | 15.44 ± 1.49 [a] | 14.48 ± 0.39 [ab] | 5.21 |
| Hemicellulose (%) | 16.63 ± 2.37 [a] | 16.18 ± 1.84 [a] | 17.01 ± 0.86 [a] | 20.40 ± 2.68 [a] | 10.95 |
| Lignin (%) | 4.60 ± 0.55 [b] | 5.97 ± 0.03 [a] | 6.03 ± 0.49 [a] | 5.48 ± 0.37 [ab] | 6.83 |
| **Colour** | | | | | |
| L | 47.71 ± 0.71 [a] | 44.84 ± 1.05 [b] | 42.81 ± 1.04 [b] | 43.72 ± 1.29 [b] | 2.30 |
| a | 29.92 ± 0.37 [a] | 24.78 ± 0.33 [b] | 24.42 ± 0.64 [b] | 25.07 ± 0.08 [b] | 1.38 |
| b | 73.76 ± 0.39 [a] | 71.71 ± 0.45 [b] | 70.46 ± 0.43 [c] | 70.66 ± 0.43 [bc] | 0.59 |

Values are given as mean ± standard deviation (n = 3). Means in the same row having different superscripts (a, b, c, d) are significantly different ($p < 0.05$). UCAM: unfermented camelina meal; ASCAM: *Aspergillus sojae* fermented camelina meal; AFCAM: *Aspergillus ficuum* fermented camelina meal; COCAM: co-culture fermented camelina meal; L: lightness-darkness; a: greenness-redness; b: yellowness-blueness; CV: coefficient of variation.

*3.3. Fibre Fractions*

There were differences ($p < 0.05$) in the fibre fractions of ADF, cellulose and lignin of the meals (Table 2). The ADF content of 16.86% in UCAM increased ($p < 0.05$) to 20.76% in ASCAM, 21.47% in AFCAM and 19.96% in COCAM. The cellulose content of ASCAM (14.79%), AFCAM (15.44%) and COCAM (14.48%) were higher ($p < 0.05$) than that of UCAM (12.25%). The lignin content of ASCAM (5.97%), AFCAM (6.03%) and COCAM (5.48%) were higher ($p < 0.05$) than that of UCAM (4.60%). In an earlier report, it was shown that fermentation increased the fibre fractions of soy protein [53]. The observed increase in ADF, cellulose and lignin contents may be attributed to the buildup of acid, alkaline and/or neutral detergent insoluble substances during SSF [17,46,54].

*3.4. In Vitro Enzyme Protein Digestion (IVPD)*

There was a significant difference ($p < 0.05$) in the in vitro enzyme protein digestion of the camelina meals (Figure 1). The IVPD of 56.14% in UCAM reduced to 40.92%, 41.54% and 29.86% in ASCAM, AFCAM and COCAM, respectively. In contrast to the present result, an increase in IVPD of soybean flour fermented with a combination of LAB strains was previously demonstrated [55]. The unanticipated decrease in IVPD may be due to the protein being locked within the fibre matrix, thus, reducing the hydrolytic action of the enzymes [17,46]. In addition, partial protein denaturation during drying may have lowered protein solubility [17,46], thus, resulting in a reduced IVPD.

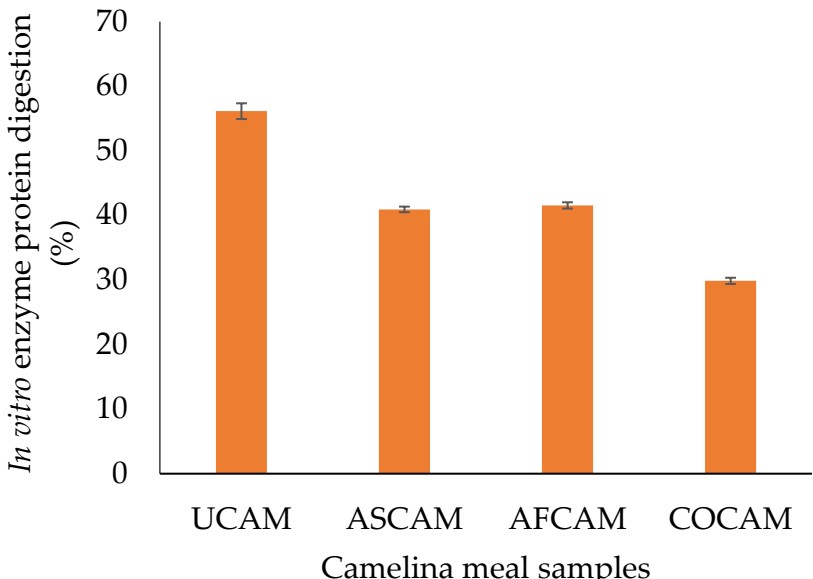

**Figure 1.** In vitro enzyme protein digestion of unfermented and solid-state fermented camelina meals. UCAM: unfermented camelina meal; ASCAM: *Aspergillus sojae* fermented camelina meal; AFCAM: *Aspergillus ficuum* fermented camelina meal; COCAM: co-culture fermented camelina meal.

*3.5. Total Glucosinolates, Total Phenolic Contents and Phytic Acid*

Total glucosinolates and total phenolic contents of the camelina meals are presented in Table 3. The total glucosinolates (26.16–30.35 µmol/g) in the present study, although not significantly different ($p > 0.05$) among treatments, were lower than 34.43 µmol/g previously reported [6]. This may be due to variations in sources of camelina meal and processing conditions. The total phenolic contents (2.52–2.64 mg GAE/g DM) observed in the present study were comparable among all treatments and fall within the range of 2.41–2.97 mg GAE/g DM reported for solid-state fermented canola meal [46]. In Figure 2, the phytic acid content of UCAM (27.48 mg/g) significantly reduced ($p < 0.05$) to lower levels of 22.39 mg/g (ASCAM), 16.72 mg/g (AFCAM) and 18.98 mg/g (COCAM), although, the phytic acid content in AFCAM did not differ ($p > 0.05$) from COCAM. The reduction in phytic acid content of camelina meal [16,56] and canola meal [20,44,46] during fermentation have been previously reported. This may be attributed to the effect of SSF in activating native phytase capable of hydrolyzing non-soluble organic complexes with minerals or a possible phytate-degrading ability of the *Aspergillus* strains [17,46]. It has been documented that the breakdown of phytic acid is pH dependent [57] and the optimal pH for most phytases is often between 4.0–6.0 [58,59]. It is also possible that a reduction in the pH of the solid-state fermented meals may have activated the phytase produced by the fungi, thereby, lowering the phytic acid contents [17].

**Table 3.** Total glucosinolates and total phenolic contents of unfermented and solid-state fermented camelina meals.

| Parameters | Total Glucosinolates (µmol SE/g DM) | Total Phenolic Contents (mg GAE/g DM) |
|---|---|---|
| UCAM | 30.35 ± 2.33 [a] | 2.57 ± 0.17 [a] |
| ASCAM | 26.48 ± 0.58 [a] | 2.64 ± 0.27 [a] |
| AFCAM | 28.80 ± 5.12 [a] | 2.64 ± 0.17 [a] |
| COCAM | 26.16 ± 0.21 [a] | 2.52 ± 0.10 [a] |
| CV (%) | 7.11 | 6.81 |

Values are given as mean ± standard deviation (n = 3). Means in the same column having different superscripts (a) are significantly different ($p < 0.05$). UCAM: unfermented camelina meal; ASCAM: *Aspergillus sojae* fermented camelina meal; AFCAM: *Aspergillus ficuum* fermented camelina meal; COCAM: co-culture fermented camelina meal; CV: coefficient of variation.

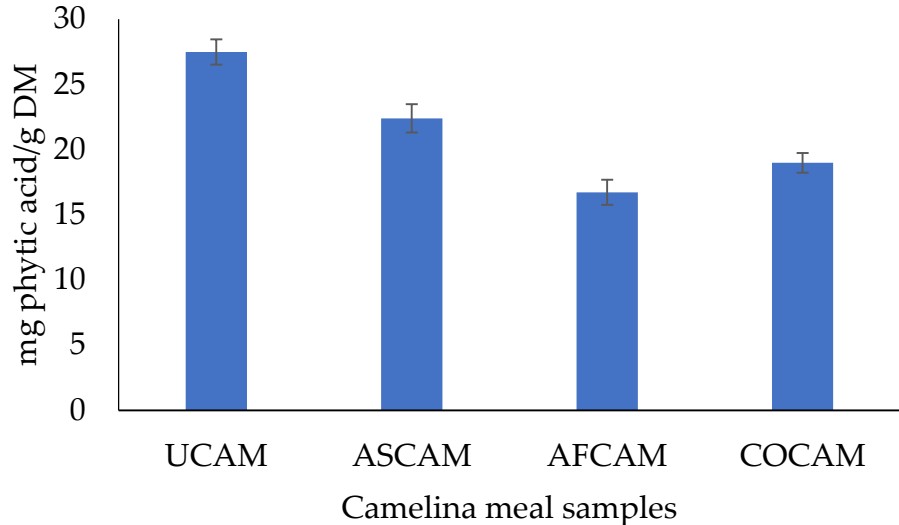

**Figure 2.** Phytic acid content of unfermented and solid-state fermented camelina meals. UCAM: unfermented camelina meal; ASCAM: *Aspergillus sojae* fermented camelina meal; AFCAM: *Aspergillus ficuum* fermented camelina meal; COCAM: co-culture fermented camelina meal.

### 3.6. Microbiological Composition

The microbiological composition of the camelina meals are presented in Table 4. After sterilization, all samples had no microbiological counts (data not shown). At the end of SSF, the estimated microbiological counts in ASCAM, AFCAM and COCAM were generally higher ($p < 0.05$) than that of UCAM, except for *Salmonella* that was not detected. Total bacteria in UCAM (2.00 log CFU/g) was significantly lower ($p < 0.05$) than in ASCAM (5.28 log CFU/g), AFCAM (4.76 log CFU/g) and COCAM (3.00 log CFU/g). The increase in total bacterial counts may be attributed to biochemical processes and changes in pH within the fermenting substrates [60]. The lactic acid bacteria (LAB) counts in UCAM (1.10 log CFU/g) were significantly lower ($p < 0.05$) to those of ASCAM (3.37 log CFU/g), AFCAM (4.50 log CFU/g) and COCAM (3.93 log CFU/g). Increase in LAB counts during lupin fermentation using *Candida utilis* has been previously reported [61]. The increase in LAB counts may be due to a reduction in pH as acidic conditions can encourage the growth of LAB [17,46,61]. Mold in UCAM (1.70 log CFU/g) was significantly lower ($p < 0.05$) than in ASCAM (4.30 log CFU/g), AFCAM (3.78 log CFU/g) and COCAM (2.70 log CFU/g). It is possible that increase in mold count may be due to the duration and favourable conditions of SSF encouraging their growth [17]. Yeast counts in UCAM (1.50 log CFU/g) were significantly lower ($p < 0.05$) than in ASCAM (3.15 log CFU/g), AFCAM (3.48 log CFU/g) and COCAM (3.52 log CFU/g). The increase in yeast count may be attributed to a reduction in the pH of the fermenting medium as yeast can tolerate low pH conditions [61].

**Table 4.** Microbiological composition of unfermented and solid-state fermented camelina meals.

| Parameters | UCAM | ASCAM | AFCAM | COCAM |
| --- | --- | --- | --- | --- |
| Total bacteria (log CFU/g) | 2.00 ± 0.27 [c] | 5.28 ± 0.16 [a] | 4.76 ± 0.21 [a] | 3.00 ± 0.52 [b] |
| Lactic acid bacteria (log CFU/g) | 1.10 ± 0.33 [c] | 3.37 ± 0.28 [b] | 4.50 ± 0.50 [a] | 3.93 ± 0.40 [ab] |
| *Salmonella* (/25 g) | ND | ND | ND | ND |
| Molds (log CFU/g) | 1.70 ± 0.50 [b] | 4.30 ± 0.20 [a] | 3.78 ± 0.33 [a] | 2.70 ± 0.22 [b] |
| Yeasts (log CFU/g) | 1.50 ± 0.32 [b] | 3.15 ± 0.51 [a] | 3.48 ± 0.19 [a] | 3.52 ± 0.25 [a] |

Values are given as mean ± standard deviation (n = 3). Means in the same row having different superscripts (a, b, c) are significantly different ($p < 0.05$). UCAM: unfermented camelina meal; ASCAM: *Aspergillus sojae* fermented camelina meal; AFCAM: *Aspergillus ficuum* fermented camelina meal; COCAM: co-culture fermented camelina meal; CFU: colony forming units; ND: not detected.

*3.7. SDS-PAGE Analysis*

The changes in the electrophoretic patterns was determined by SDS-PAGE (Figure 3). The difference in protein molecular weights was quantified by a densitometric analysis using Image Lab version 5.0 (Bio-Rad Laboratories, Hercules, CA, USA). SDS-PAGE pattern showed that SSF influenced the protein molecular distribution of the CAM samples. The two major storage proteins in UCAM (lane 1) which are cruciferin (12S globulin) and napin (2S albumin) have been previously documented [62,63]. In the present study, the approximate molecular weights of cruciferin (12S globulin) subunits were 95.9, 75, 63.8, 48.5, 41.6, 35.6, 33 and 30 kDa and napin (2S albumin) subunits were 24.7, 19.7, 11.1, 9 and 5.6 kDa. However, the large-sized proteins were found to have disappeared in ASCAM, AFCAM and COCAM (lanes 3, 4 and 5) and more small-sized proteins or peptides (less than 33.6 kDa) were observed. It is possible that large and mid-sized proteins of CAM were hydrolyzed into small molecular weight proteins by proteolytic enzymes produced by the *Aspergillus* strains during SSF [17,45,46]. In an earlier study, the molecular weight of the main protein fractions in unfermented rapeseed meal were 72, 55 and 37 kDa [45]. SSF of rapeseed meal using *Aspergillus niger* increased the amount of small-size peptides and significantly decreased large-size peptides [45].

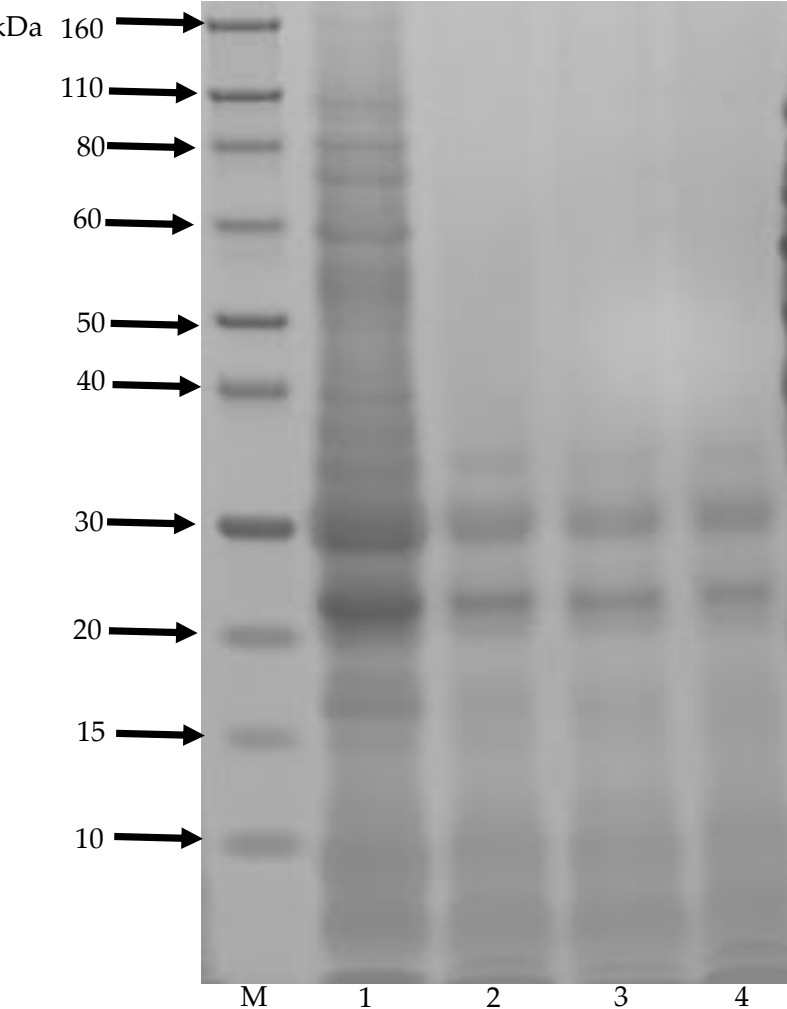

**Figure 3.** SDS-PAGE patterns of unfermented and solid-state fermented camelina proteins. Lane M: molecular weight marker (kDa); lane 1: unfermented camelina meal (UCAM); lane 2: *Aspergillus sojae* fermented camelina meal (ASCAM); lane 3: *Aspergillus ficuum* fermented camelina meal (AFCAM); lane 4: co-culture fermented camelina meal (COCAM).

## 3.8. Functional Properties

The effect of SSF on functional properties of CAM samples is presented in Table 5. The bulk density of 0.53 g/mL (in UCAM) significantly increased ($p < 0.05$) to 0.56 g/mL (ASCAM), 0.56 g/mL (AFCAM) and 0.57 g/mL (COCAM), although the bulk density values were not significantly different ($p > 0.05$) among the solid-state fermented meals. The result is consistent with a previous report on increased bulk density after the fermentation of sandbox seeds flour [64]. Increased bulk density suggests better product packaging as higher quantity may be packed within a constant volume [65], thus an advantage for the fermented meals. The water absorption capacity of UCAM (650%) significantly reduced ($p < 0.05$) to 395% (ASCAM), 390% (AFCAM) and 385% (COCAM), although the water absorption capacity values were not significantly ($p > 0.05$) different among the solid-state fermented meals. This result aligns with a previous report on decreased water absorption capacity after the fermentation of pigeon pea seed flour [66]. The reduction in water absorption capacity may be due to reduced hydrophilic groups which bind water molecules during SSF [67] indicating the potential use of the fermented meals for producing thin gruels in food formulations. The oil absorption capacity varied ($p < 0.05$) among the meals ranging from 90% (ASCAM), 100% (UCAM), 100% (COCAM) and 110% (AFCAM), with ASCAM having the lowest ($p < 0.05$) oil absorption capacity value. A reduction in oil absorption capacity of fermented mahogany bean flour has also been previously shown [68]. In contrast, red bean flour showed increase in oil absorption capacity after SSF [69]. Oil absorption capacity of food proteins depend on essential factors such as amino acid composition, protein structure and surface polarity or hydrophobicity [70]. The increase in the oil absorption capacity of AFCAM may be indications of increased availability of hydrophobic amino acids by opening the non-polar residues out from the interior protein molecules during SSF [69]. This suggests AFCAM can serve as a flavour retainer in food system, for instance in sausage production where optimal oil absorption is preferred. The swelling index of 3.19% (in UCAM) significantly increased ($p < 0.05$) to 3.99% (in AFCAM) but reduced to 2.79% (in ASCAM) and 2.93% (COCAM). A reduction in the swelling index of fermented brown rice flour has also been previously reported [71]. The decrease in swelling index of ASCAM and COCAM may be due to pre-denaturation of protein during SSF. The increase in swelling index of AFCAM may be indications of high water affinity [72], which might ultimately produce a good value-added product. The swelling capacity of 3.90 (UCAM) significantly increased ($p < 0.05$) to 4.78 (ASCAM), 4.76 (AFCAM) and 4.52 (COCAM), although the swelling capacity values were not significantly different ($p > 0.05$) among the fermented meals. Increase in the swelling capacity in the present study is consistent with a previous report of increased swelling capacity of fermented pearl millet flour [72]. In general, the increase in swelling capacities of the solid-state fermented camelina meals indicate their potential in developing bakery products since improved swelling capacities are useful indicators that they can be incorporated into aqueous food formulations mostly during the preparation of dough.

**Table 5.** Functional properties of unfermented and solid-state fermented camelina meals.

| Parameters | UCAM | ASCAM | AFCAM | COCAM | CV (%) |
|---|---|---|---|---|---|
| Bulk density (g/mL) | 0.53 ± 0.01 [b] | 0.56 ± 0.01 [a] | 0.56 ± 0.01 [a] | 0.57 ± 0.00 [a] | 1.36 |
| WAC (%) | 650 ± 110.00 [a] | 395 ± 25.00 [b] | 390 ± 0.00 [b] | 385 ± 55.00 [b] | 9.38 |
| OAC (%) | 100.00 ± 0.00 [ab] | 90.00 ± 10.00 [b] | 110.00 ± 10.00 [a] | 100.00 ± 0.00 [ab] | 5.05 |
| Swelling index | 3.19 ± 0.00 [ab] | 2.79 ± 0.13 [b] | 3.99 ± 0.67 [a] | 2.93 ± 0.00 [b] | 5.36 |
| Swelling capacity | 3.90 ± 0.10 [b] | 4.78 ± 0.22 [a] | 4.76 ± 0.17 [a] | 4.52 ± 0.04 [a] | 2.91 |
| Dispersibility (%) | 90.87 ± 0.18 [a] | 90.78 ± 0.35 [a] | 90.42 ± 0.09 [a] | 90.82 ± 0.31 [a] | 0.26 |

Values are given as mean ± standard deviation (n = 3). Means in the same column having different superscripts (a, b) are significantly different ($p < 0.05$). UCAM: unfermented camelina meal; ASCAM: *Aspergillus sojae* fermented camelina meal; AFCAM: *Aspergillus ficuum* fermented camelina meal; COCAM: co-culture fermented camelina meal; WAC: Water absorption capacity; OAC: Oil absorption capacity; CV: coefficient of variation.

## 4. Conclusions

The use of food grade *Aspergillus* fungi (*Aspergillus sojae*, *Aspergillus ficuum* and their co-cultures) influenced the physicochemical composition and microbiological properties of camelina meal. SSF improved the nutritional value and reduced the phytic acid content of camelina meal. In addition, SSF improved the functional properties by increasing the bulk density and swelling capacity, and reducing the water absorption capacity). *Aspergillus ficuum* fermented camelina meal had the highest oil absorption capacity and swelling index among all samples. Based on the results of this study, camelina meal can serve as a viable substrate for SSF and has the possibility of creating value-added novel products with functional application in food and feed production.

**Author Contributions:** Conceptualization: O.O.O.; R.M.; X.L. and Y.S.; Data curation: O.O.O.; Formal analysis: O.O.O.; W.C.F. and Y.S.; Investigation: O.O.O.; W.C.F.; R.M.; X.L. and Y.S.; Methodology: O.O.O.; W.C.F.; R.M.; X.L. and Y.S.; Supervision: R.M.; X.L. and Y.S.; Writing—review and editing: O.O.O.; R.M.; X.L. and Y.S. All authors have read and agreed to the published version of the manuscript.

**Funding:** This research received no external funding.

**Acknowledgments:** The authors are sincerely grateful for the support of the University of Queensland and Queensland Department of Agriculture and Fisheries, Brisbane, Australia. The support through the Research Training Program Scholarship provided to Oladapo Oluwaseye Olukomaiya during his PhD study at the University of Queensland, Brisbane is gratefully acknowledged.

**Conflicts of Interest:** The authors declare no conflict of interest.

## Abbreviations

UCAM, unfermented camelina meal; ASCAM, *Aspergillus sojae* fermented camelina meal; AFCAM, *Aspergillus ficuum* fermented camelina meal; COCAM, co-culture fermented camelina meal; SSF, solid-state fermentation.

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
