# Peer review of "Physicochemical, Microbiological and Functional Properties of Camelina Meal Fermented in Solid-State Using Food Grade Aspergillus Fungi"

_fermentation, doi:10.3390/fermentation6020044_

Round 1

Reviewer 1 Report

The issue of innovative food and feed components is very interesting and important for the circular economy and sustainable development.

Manuscript is prepared correctly. The methodology is adequate to the research goal and the results are well presented and discused.

I have one remark only - section 3. sould be named: Results and discusion.

Author Response

Editorial board member’s comments

Value of your results. Although there were improvements on the concentration
of for example protein and starch, those seem to be not to high. An
explanation regarding the value that those improvements have would enhance the
manuscript.

Author’s response: Thank you for the comments. As regards the value of our results, statistical analysis was done, with a significance level of p < 0.05. The experiments were conducted in triplicate and all measurements were done in triplicate. Although the concentrations were not high, the mean differences among treatments have been shown after conducting the statistical analysis and the results have been discussed in the manuscript.

Additionally, at this point there is only one figure in the whole manuscript
and several tables. I recommend to include a couple of figures more in order
to make the manuscript more attractive for the readers.

Author’s response: Thank you for the comments. We have added two more figures as suggested in the revised text (pls see Figures 1 and 2).

Reviewer #1

The issue of innovative food and feed components is very interesting and important for the circular economy and sustainable development.

Author’s response: Thank you for your compliment and understanding of our work. The invaluable comments are appreciated.

Manuscript is prepared correctly. The methodology is adequate to the research goal and the results are well presented and discussed.

Author’s response: Thank you for your compliment and understanding of our work. The invaluable comments are appreciated.

I have one remark only - section 3. should be named: Results and discussion.

Author’s response: Thank you for the invaluable comments. The suggestion has been effected in the revised text (pls see line 229).

Reviewer 2 Report

*Overview and general recommendation
The authors aimed to assess different Aspergillus species and a co-culture in solid-state fermentation of camelina meal. I found the manuscript inside the scope of the journal, but some paragraphs must be revised in the style of English or cited literature. The authors used in most of the paragraphs statistical analyses, showing support to the results, but some of the paragraphs that included discussion are lacking literature supporting the affirmations. Therefore, I recommend accepting after minor revisions. In the next lines, I comment on some concerns.

Line 229: Did you perform all the assumptions of normality and homogeneity of variances? This is very important to highlight because all your results are supported on the statistical analyses. 

Lines 241-242, 294-297. Some paragraphs in the discussion are lacking literature that supports the affirmations. Please revise all the paragraphs in the same condition and add references to the manuscript.

Line 300. Please revise the text "in Table 3. In Table 3,". It sounds redundant.

Lines 331, 334, 377, others. In some parts of the manuscript appears the text "significantly (p > 0.05) different" or "significantly (p < 0.05) lower"... I recommend writing the P-value after the adjective/adverb. 

Lines 350-353. Did you perform any experiment to determine the concentration of total proteins in each lane? Did you standardize the concentration of proteins?

Line 385. Please standardize the citation. In the same sense, In many parts of the text appeared the citation as "author et al. [22]". My suggestion is to fix the redaction in order to reduce the mixed style of citations. 

Author Response

Reviewer #2

Overview and general recommendation
The authors aimed to assess different Aspergillus species and a co-culture in solid-state fermentation of camelina meal. I found the manuscript inside the scope of the journal, but some paragraphs must be revised in the style of English or cited literature.

Author’s response: Thank you for the invaluable comments and suggestions. All queries have been responded and corrections have been made as highlighted in red in the revised text. We have improved the paragraphs by adding relevant references as suggested in the the revised text. (pls see lines 246, 248, 283, 296, 303, 304, 322, 325, 349, 352, 355, 372, 374, 375, 411)

The authors used in most of the paragraphs statistical analyses, showing support to the results, but some of the paragraphs that included discussion are lacking literature supporting the affirmations. Therefore, I recommend accepting after minor revisions. In the next lines, I comment on some concerns.

Author’s response: Thank you for your suggestion. We have added relevant references in support of the affirmations made in the revised text (pls see lines 246, 248, 283, 296, 303, 304, 322, 325, 349, 352, 355, 372, 374, 375, 411)

Line 229: Did you perform all the assumptions of normality and homogeneity of variances? This is very important to highlight because all your results are supported on the statistical analyses. 

Author’s response: Thank you for your question. Yes, we performed all the assumptions of normality and homogeneity of variances during the statistical analyses and this has been highlighted in the revised text (pls see line 228)

Lines 241-242, 294-297. Some paragraphs in the discussion are lacking literature that supports the affirmations. Please revise all the paragraphs in the same condition and add references to the manuscript.

Author’s response: Thank you for your suggestion. We have added relevant references in support of the affirmations made in the revised text (pls see lines 246, 248, 283, 292, 303, 304, 333, 343, 354, 360, 377,378, 384, 388, 389, 398, 403)

Line 300. Please revise the text "in Table 3. In Table 3,". It sounds redundant.

Author’s response: Thank you for your suggestion. The text has been revised in the revised text (pls see line 314)

Lines 331, 334, 377, others. In some parts of the manuscript appears the text "significantly (p > 0.05) different" or "significantly (p < 0.05) lower"... I recommend writing the P-value after the adjective/adverb. 

Author’s response: Thank you for your comment. The suggestions have been effected by indicating the P-value after the adjective in the revised text.

Lines 350-353. Did you perform any experiment to determine the concentration of total proteins in each lane? Did you standardize the concentration of proteins?

Author’s response: Thank you for your questions. Yes, we determined the concentration of total protein in each lane and the concentration of the proteins were standardized. After extraction, BCA assay was done to quantify the total protein content (protein concentration) of the samples. After quantification, equal amount of total protein was carefully loaded in all the wells to ensure the consistence of loading the same quantity of protein samples.

Line 385. Please standardize the citation. In the same sense, In many parts of the text appeared the citation as "author et al. [22]". My suggestion is to fix the redaction in order to reduce the mixed style of citations. 

Author’s response: Thank you for your comments. The suggestion has been effected to ensure a uniform style of citations within the manuscript.

Reviewer 3 Report

The paper is so good for publication in this journal.

Author Response

Reviewer #3: The paper is so good for publication in this journal.

Author’s response: Thank you for your comments and understanding of our work. The invaluable comments are appreciated.